# Large-scale analysis of SARS-CoV-2 spike-glycoprotein mutants demonstrates the need for continuous screening of virus isolates

Barbara Schrörs[1], Pablo Riesgo-Ferreiro[1‡], Patrick Sorn[1‡], Ranganath Gudimella[1☯], Thomas Bukur[1☯], Thomas Rösler[1], Martin Löwer[1]*, Ugur Sahin[1,2]*

1 Biomarker Discovery Center, Translationale Onkologie an der Universitätsmedizin der Johannes Gutenberg-Universität Mainz Gemeinnützige GmbH, Mainz, Rhineland-Palantinate, Germany, 2 CEO, BioNTech SE, Mainz, Rhineland-Palantinate, Germany

☯ These authors contributed equally to this work.
‡ These authors also contributed equally to this work.
* martin.loewer@tron-mainz.de (ML); sahin@uni-mainz.de (US)

## Abstract

Due to the widespread of the COVID-19 pandemic, the SARS-CoV-2 genome is evolving in diverse human populations. Several studies already reported different strains and an increase in the mutation rate. Particularly, mutations in SARS-CoV-2 spike-glycoprotein are of great interest as it mediates infection in human and recently approved mRNA vaccines are designed to induce immune responses against it. We analyzed 1,036,030 SARS-CoV-2 genome assemblies and 30,806 NGS datasets from GISAID and European Nucleotide Archive (ENA) focusing on non-synonymous mutations in the spike protein. Only around 2.5% of the samples contained the wild-type spike protein with no variation from the reference. Among the spike protein mutants, we confirmed a low mutation rate exhibiting less than 10 non-synonymous mutations in 99.6% of the analyzed sequences, but the mean and median number of spike protein mutations per sample increased over time. 5,472 distinct variants were found in total. The majority of the observed variants were recurrent, but only 21 and 14 recurrent variants were found in at least 1% of the mutant genome assemblies and NGS samples, respectively. Further, we found high-confidence subclonal variants in about 2.6% of the NGS data sets with mutant spike protein, which might indicate co-infection with various SARS-CoV-2 strains and/or intra-host evolution. Lastly, some variants might have an effect on antibody binding or T-cell recognition. These findings demonstrate the continuous importance of monitoring SARS-CoV-2 sequences for an early detection of variants that require adaptations in preventive and therapeutic strategies.

## Introduction

Since the first report of the severe acute respiratory syndrome coronavirus-2 (SARS-CoV-2) outbreak [1, 2], it has transformed into a global pandemic infecting and threatening death for millions of people all over the globe. By July 9, 2021, the World Health Organization (WHO)

**Data Availability Statement:** All relevant data are within the manuscript and its Supporting

Information files. Raw data was obtained from public resources (SRA, GSAID).

**Funding:** The study supported by BioNTech SE, Mainz, Germany. The funder provided support in the form of salary for author U.S., but did not have any additional role in the study design, data collection and analysis, decision to publish, or preparation of the manuscript. The specific roles of this author is articulated in the 'author contributions' section. In addition, the other authors are employees of the non-profit company TRON gGmbH and are supported in form of salary. TRON gGmbH did not have any additional role in the study design, data collection and analysis, decision to publish, or preparation of the manuscript.

**Competing interests:** Author U.S. is co-founder, shareholder and CEO at BioNTech SE. This does not alter our adherence to PLOS ONE policies on sharing data and materials. The remaining authors declare that the research was conducted in the absence of any commercial or financial relationships that could be construed as a potential conflict of interest

reported 185,291,530 confirmed cases and 4,010,834 deaths caused by the SARS-CoV-2 outbreak [3]. After the approval of SARS-CoV-2 vaccines which are designed to invoke immune responses against the spike-glycoprotein (spike protein), it becomes necessary to track the mutations in spike protein and study their relevance for current and upcoming vaccines. Also the recently approved neutralizing antibody bamlanivimab targets the spike protein of SARS-CoV-2 [4].

Subunits of the spike protein are valuable targets for vaccine design as the protein is responsible for viral binding and entry to host cells [5, 6]. The spike protein consists of the N-terminal S1 and the C-terminal S2 subunits; the receptor-binding domain (RBD) in the S1 subunit binds to a receptor on the host cell surface and the S2 subunit fuses viral and host membranes [7]. The receptor binding domain (RBD) of the SARS-CoV-2 spike protein recognizes human angiotensin-converting enzyme 2 (ACE2) as its entry receptor, similar to SARS-CoV [8]. Interacting residues of the SARS-CoV-2 RBD with human ACE2 are highly conserved or share similar side chain properties with the SARS-CoV RBD [9]. In addition, the SARS-CoV-2 RBD shows significantly higher binding affinity to ACE2 receptor compared to the SARS-CoV RBD. In order to repress the infection, blocking the RBD binding was effective in ACE2-expressing cells [5]. Among the interacting sites in the SARS-CoV-2 RBD, particularly the amino acid residues L455, F486, Q493, S494, N501, and Y505 provide critical interactions with human ACE2 [10]. These interacting residues vary due to natural selection in SARS-CoV-2 and other related coronaviruses [11]. Similarly, worldwide SARS-CoV-2 genomic data shows ten RBD mutations which were caused due to natural selection by circulating among the human population [12]. RBD mutations particularly at N501 may enhance the binding affinity between SARS-CoV-2 and human ACE2 significantly, improving viral infectivity and pathogenicity [10].

It is reported that continuous evolution of SARS-CoV-2 among the global population results into six major subtypes which involve the recurrent D614G mutation of the spike protein [13]. Further, spread of such recurrent mutations within sub-populations might affect the severity of disease emergence and change the trajectory of the pandemic. Studies also report high intra-host diversity caused by low frequency subclonal mutations within a specific cohort [14]. It is evident that changes in the SARS-CoV-2 genome over time might show new mutations which might influence the development efforts of interventional strategies. The variability of epitopes of the RBD might hamper the development and use of neutralizing antibodies for cross-protective activities against mutant strains [15]. Mutational variants of the spike protein might as well lead to escape variants with respect to pre-existing cross-reactive CD4+ T cell responses [16] or long-term protection from re-infection through T cell memory. Hence, there is a necessity of constant monitoring of the rapidly changing mutation rates in the spike protein in SARS-CoV-2, which could have significant impact on virus infection, transmissibility and pathogenicity in the current pandemic.

In this study, we gathered 1,036,030 genomic assemblies and 30,806 NGS datasets to detect non-synonymous spike protein mutations and infer their frequency within a given sample and the effect on potential antibody binding sites and known T cell epitopes.

## Methods

### SARS-CoV-2 assemblies

SARS-CoV-2 assemblies from human hosts were downloaded on April 13th, 2021 from GISAID (nucleotide sequences; [17]). Two samples were excluded for having seemingly wrong dates, 2012-03-21 and 2017-11-21. 2,014 additional samples were excluded as the assemblies were shorter than 1,000 bp. Unfortunately, all 98 samples from Ivory Coast failed to be loaded

due to a technical problem, this issue will be amended in subsequent analyses. Pairwise alignments to the reference genomic sequence (MN908947.3) were performed using the Python package Biopython (version 1.79). Global alignment options were (extend_gap_score = -0.1, open_gap_score = -3,_mismatch_score = 1,_match_score = 2). Variants were called using the global alignment results. Any variant containing an N or any ambiguous IUPAC code (ie: (ie: R, Y, S, W, K, M, B, D, H, V)) was filtered out. With the previous procedure, we obtained an average of 22.425SNVs per sample respectively. We further excluded all samples with a number of variants greater than 75th percentile plus three times the interquartile range for each variant type separately. This resulted in 120 samples excluded due to an extremely high number of SNVs. The variants were subsequently normalized following the procedure described in Tan et al. 2015 [18] using vt (version 0.57721) and BCFtools (version 1.12). Finally, variants were annotated with SnpEff (version 5.0) [19]. This analysis workflow was implemented in the Nextflow framework and open sourced under the MIT license as the CoVigator NGS pipeline (https://github.com/TRON-Bioinformatics/covigator-ngs-pipeline) [20].

## NGS data processing

All available NGS data for SARS-CoV-2 was downloaded on June 11th, 2021 from the European Nucleotide Archive (ENA) application programming interface (API) (https://www.ebi.ac.uk/ena/portal/api/; [21] and filtered for whole genome FASTQ data from Illumina instruments with a human sample background. Data were aligned to the reference MN908947.3 [2].

Short-read whole genome sequencing data were aligned with bwa (version 0.7.17) mem [22]. Output files in SAM format were sorted and converted to their binary form (BAM) using SAMtools (version 0.1.16) [23]. Variants were retrieved from the alignment files using BCFtools (version 1.9) mpileup (http://samtools.github.io/bcftools/) with the options to recalculate per-base alignment quality on the fly, disabling the maximum per-file depth, and retention of anomalous read pairs. Variants in gene gp02 (i.e. S gene) were annotated using SNPeff (version 4.3t) "ann" [19].

## Filtering subclonal variants

NGS variants were filtered with at least 30 reads coverage and a fraction of supporting reads of at least 0.1 and less than 0.95 to identify high-confidence sub-clonal mutations [24].

## Published SARS-CoV-2 T-cell epitopes and HLA binding prediction

SARS-CoV-2 antigens reported by Snyder et al. [25] where downloaded from https://clients.adaptivebiotech.com/pub/covid-2020 on 17NOV2020 (MIRA release 002.1).

343 spike protein epitopes with positive T cell response were extracted from the IEDB database (https://www.iedb.org/, accessed June 21th, 2021).

HLA binding prediction was done with netMHCpan, version 4.1 [26].

## Results

## SARS-CoV-2 spike protein mutational profile from genome assemblies and NGS data

First, we determined the number of non-synonymous mutations in the spike protein per sample (for geographic background of the collected samples, see S1 Fig). Of the 1,036,030 analyzed genome assemblies and 30,806 NGS data sets, only 2.5% (26,746 samples) contained the wild type (WT) spike protein (Fig 1A). Samples of mutant viruses exhibited only few mutations in the spike protein with less than ten mutations for all but 4,193 sequences. However, the mean

**A**

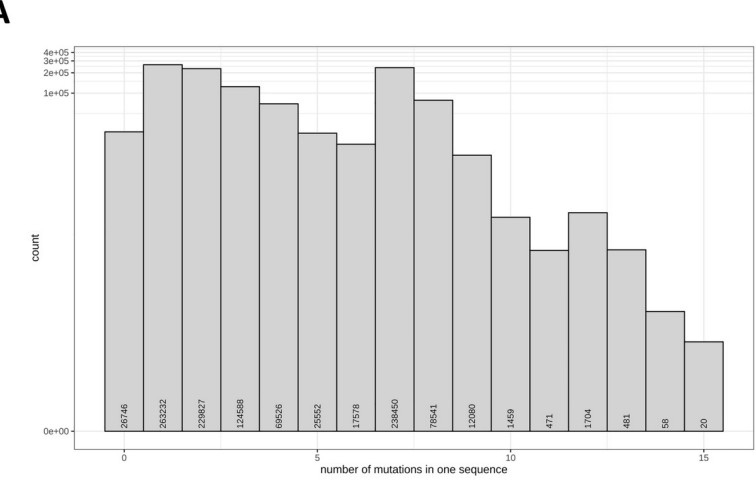

**B**

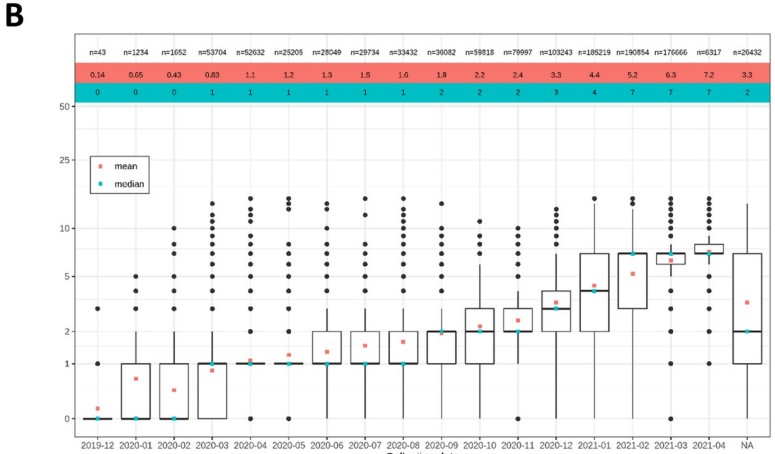

**Fig 1. Most of the analyzed SARS-CoV-2 sequences differ from WT spike protein, but exhibit only few non-synonymous mutations.** (A) The histogram shows the number of non-synonymous spike protein mutations detected in the analyzed samples. (B) The mean (red) and median (blue) number of mutations per spike protein sequence increased over time. The top line gives the number n of samples per month. The boxplots indicate the monthly distributions of mutations per sample.

and median number of mutations increased over time from December 2019 (mean: 0.14, median: 0) to April 2021 (mean: 7.2, median: 7; Fig 1B). Overall, we detected 5,472 distinct non-synonymous mutations in the spike protein (S1 Table).

## Recurrent variants in SARS-CoV-2 spike protein

Most of the observed variants in the assembly and NGS data sets were recurrent (Fig 2A) and only 22.4% of the variants were singular events in the combined assembly and the NGS data. The recurrent variants were distributed throughout the whole spike protein (Fig 2B and 2C). Among the recurrent variants, 21 and 14 mutations were found in at least 1% of the mutant assembly and NGS samples, respectively (labeled variants in Fig 2B and 2C). The most common mutation was D614G in both the genome assemblies (1,056,342 samples) and the NGS data (27,667 samples) located outside the RBD (positions 319–529), followed by the RBD variants Y501N in the assemblies (346,194 samples) and in the NGS data (5,987 samples). In total, 852 distinct mutations (646 recurrent) were detected in the RBD in the assemblies out of

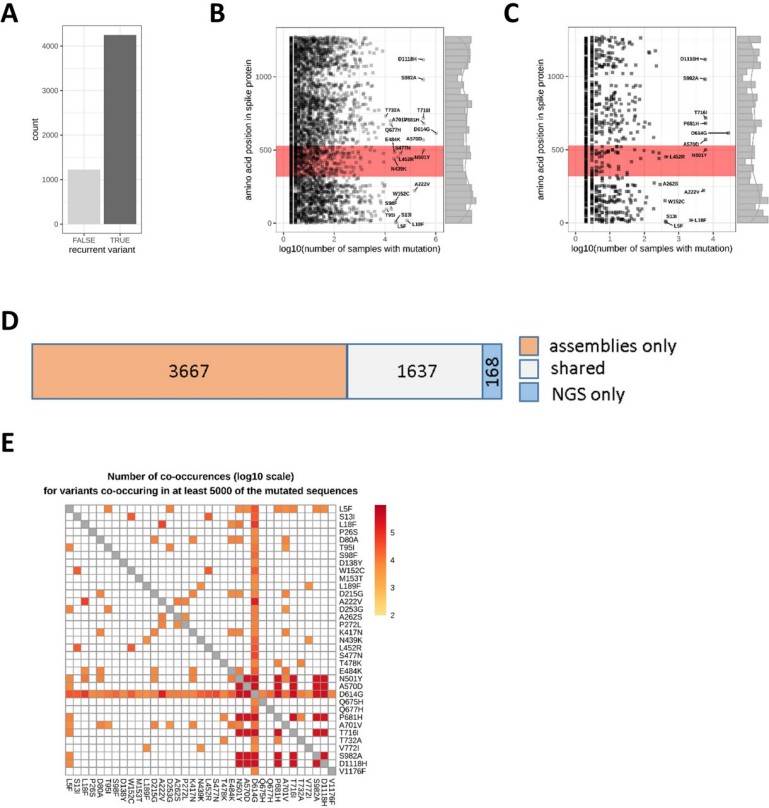

**Fig 2. Recurrent variants are found throughout the whole spike protein.** (A) Most of the detected variants were recurrent events occurring in at least two samples from the assembly or NGS data sets. (B, C) Each data point represents a distinct protein sequence mutation in the spike protein. The labels indicate the amino acid exchange for variants found in more than 1% of the assemblies (B) or NGS samples (C). The RBD is highlighted in red. (D) 1,637 variants (grey) were detected both in the assemblies and the NGS data. (E) A subset of 35 variants co-occurred in at least 5000 of the mutated spike protein sequences (assemblies and NGS data combined). For better visibility, co-occurrences in less than 5000 samples were set to 0 (white tiles).

which only 5 were common to more than 1% of the mutated assembly sequences (Fig 2B). For the NGS samples, 259 mutations in total (105 recurrent) were found in the RBD (Fig 2C) and only two were detected in at least 1% of the mutant NGS samples. Overall, 1,637 mutations were commonly found in the assembly and NGS data (Fig 2D).

Furthermore, 35 (0.64%) of the detected variants co-occurred frequently in at least 5000 of the mutated spike protein sequences when we combined assembly and NGS data (Fig 2E). Most prominent here, was the variant D614G which was found in combination with 4,066 other variants. The combination P681H/D614G was detected in 345,808 samples. The most frequent co-occurring mutations not involving D614G were P681H/T716I (324,269 samples).

## Subclonal variants

In addition, we were interested in subclonal spike protein mutations (i.e. mutations with an observed variant frequency—as derived from the NGS reads—below 100%) which might either indicate co-infection with various SARS-CoV-2 strains and/or intra-host evolution of the virus. To this end, the fraction of variant supporting reads per sample of the detected mutations was determined. Most of the variants were observed with at least 95% of the reads supporting the respective variant nucleotide (Fig 3A and 3B). However, a portion of the

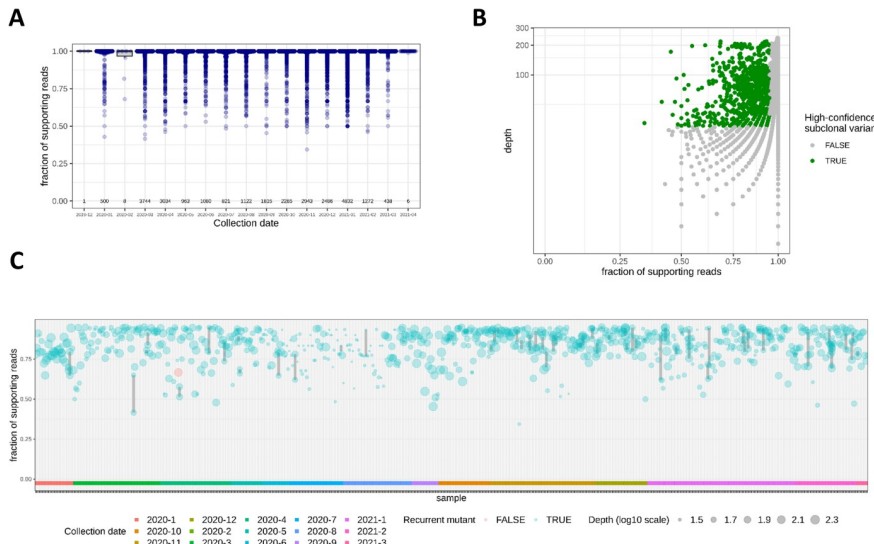

**Fig 3. Variant frequencies of spike protein mutants indicate presence of multiple SARS-CoV-2 mutants in some samples.** (A) The boxplot shows the distributions of the fraction of supporting reads of the mutations found in the NGS data. The numbers of underlying samples are indicated above the collection dates. Most of the observed variants have a variant allele frequency of $>=0.95$ and can be accounted as clonal. (B) Filtering for high-confidence subclonal variants (green) with sequencing depth $>=30$ reads and fractions of supporting reads between 0.1 and 0.95. (C) Sample-wise depiction of high-confidence subclonal events. Some of the observed subclonal variants were recurrent (blue) and only few were individual (red). The samples were ordered by collection date (see also color bar at the bottom of the plot) and point sizes indicate sequencing depth (log10 scale). Subclonal variants of the same sample are linked with grey lines. The fraction of supporting reads of variants found in the same sample differed notably in some cases.

overlapping reads pointing to subclonal events only confirmed among few mutations. Filtering for a depth of at least 30 reads and a fraction of supporting reads between 0.1 and 0.95 [24] resulted in 834 mutations observed in 732 samples (i.e. 2.59% of the NGS data sets with mutant spike protein) that could be classified as high-confident subclonal (Fig 3B). Most of these subclonal events were recurrent variants (Fig 3C). In some of the earlier samples, but also in some

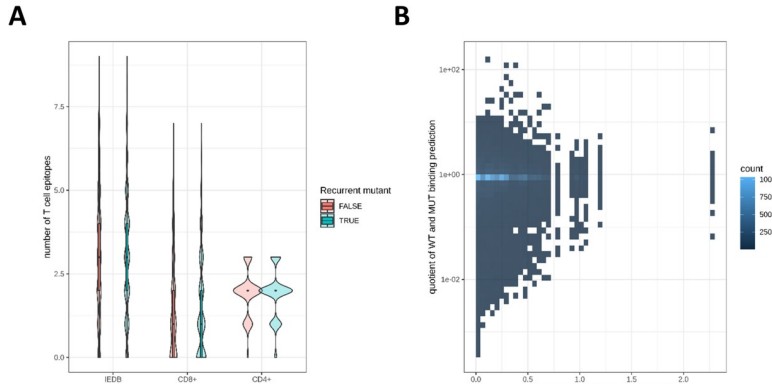

**Fig 4. Variants affect antibody and T cell target sites.** (A) The number of published T cell epitopes (listed in the IEDB or recognized by CD8+ or CD4+ T cells as reported by Snyder *et al.* [25] that are affected by recurrent or individual spike protein variants is depicted. Most of the variants hit at least one epitope. (B) Predicted binding affinity rank of wild type epitope (WT) vs. the quotient of the predicted rank of wild type and mutant epitope (MUT) depicted as heat map. Lower ranks indicate better binding. A small quotient indicates worse binding prediction for the mutant epitope.

later cases, the fractions of supporting reads within the same sample differed notably (grey lines in Fig 3C) indicating the presence of more than two spike protein versions within the same host.

## Effect of detected spike protein variants on potential antibody and T cell target sites

Next, we investigated whether the observed spike protein variants were relevant in the context of T cell recognition. 5390 (98.5%) of the 5,472 distinct variants hit at least one CD8+ or CD4 + T-cell epitope (Fig 4A) reported by Snyder et al. [25] no matter if they were recurrent or individual events and 4959 (90.6%) variants overlap with T-cell epitopes from the IEDB. In order to evaluate the qualitative impact of mutations in known T-cell epitopes, we collected 225 MHC class I epitopes with known HLA restriction from various literature sources [27–29], allowing the use of computational tools [26] to predict the difference of binding affinities of the mutant and the wild-type epitope. This computed value has been demonstrated to be predictive for immunogenicity [30]. The results for 15905 epitope-mutation pairs (Fig 4B) show a lower predicted binding affinity by 100-fold or more for 266 mutant epitopes and by 1000-fold or more for 13 mutant epitopes.

## Discussion

Our study sheds light on non-synonymous variants in the spike protein of SARS-CoV-2 in a large cohort of samples from all over the world. While most analyzed sequences vary from the reference sample from Wuhan, China, our analysis of more than one million assembly and NGS samples shows an overall low mutation burden in the SARS-CoV-2 spike protein across different host populations (Fig 1). However, the mean and median number of variants per sample increased over time. Coronaviruses have fewer mutations compared to any other RNA virus due to its inherent 3' to 5' exoribonuclease activity [31]. This suggests that the SARS-CoV-2 genome is genetically stable and the vast majority of mutations have no phenotypic effect such as virus transmissibility and virulence [32, 33]. However, mutations of critical residues in the RBD of the spike protein might increase the virus transmission ability by enhancing the interaction [34]. Furthermore, vaccines or treatments targeting the spike protein might become less efficient, if the number of variants in the spike protein increases further, as described by McCallum et al. [35].

We identified a subset of mutations from the assembly and NGS data that are recurrent variants in the spike protein. Van Dorp et al. [36] have already reported such recurrent variants in SARS-CoV-2 evolution, which is a likely phenomenon of positive selection signifying the adaption of SARS-CoV-2 in human hosts. Furthermore, most recurrent variants show no evidence in increase of viral transmission and are likely induced by host immunity through RNA editing mechanisms [37]. However, some variants might significantly influence SARS-CoV-2 transmission and infectivity. Among such variants, the non-synonymous D614G mutation has become most prevalent among several populations. We identified around 99.1% of the samples with a D614G variant, which supports a previous theory of an increasing frequency of the D614G variant in the global pandemic [34]. Studies show evidence that the D614G variant is associated with high levels of viral RNA in COVID-19 patients, suggesting a role of D614G mutations in enhancing the viral infectivity in patients [34, 38–40]. In contrast to these findings, it remains unclear whether the D614G variant makes the infections more severe or may impact vaccine design [41], as the viral load does not correlate with disease severity and the variant is not in the RBD of the spike protein, which interacts with the human ACE2 protein.

The RBD of the spike protein is a potential target for neutralizing antibodies and the variants in these regions might influence the infectivity and pathogenicity. We have identified high frequency variants in the RBD region from the assembly data, i.e. S477N, Y501N or R452L (Fig 2B and 2C). S477N occurs frequently and studies show that S477N has potential to affect the RBD stability and strengthen the binding with the human ACE2 protein [42, 43]. In our study, S477N was frequently co-occurring with D614G (Fig 2D). This combination was estimated to spread more rapidly than the D614G mutant alone [44]. Other RBD variants such as N439K and N440K also show enhanced binding affinity to the human ACE2 receptor and result in immune escape from a panel of neutralizing monoclonal antibodies [45–47]. Antibody-resistant RBD variants might affect the therapeutic potential of neutralizing monoclonal antibodies by escaping through disruption of epitopes.

However, a significant portion of the detected variants represent individual events based on what could be deduced from the available data. This indicates the necessity to further collect SARS-CoV-2 isolates and monitor newly occurring variants. Here, the combination of assembly data (which appeared to be available in a timelier manner) and NGS samples (which also contain information on the clonality of the observed variants but which might be deposited with some delay) provide a valuable resource.

Further, we identified subclonal variants with a fraction of supporting reads between 0.1 and 0.95 at a sequencing depth of more than 30 reads in 2.59% of the NGS samples with mutant spike protein (Fig 3). Subclonal variants are indicative of within-host viral diversity leading to transmission of multiple strains [24]. Low frequency variants could have been part of parallel evolution, where the same mutation rises to detectable frequencies in different lineages and it is observed as part of SARS-CoV-2 virus adaptation [48]. Further, recurrent mutations might point to co-infection with multiple strains. Sample-specific variants in turn might rather indicate that the mutation occurred after infection within the host. This viral diversity within the host might prevent complete clearance after treatment and thus might lead to the development of resistant strains. Also, subclonal variants should be considered for vaccine design as these might represent the next generation of the virus.

The analyzed data sets also showed that a notable portion of the individual and recurrent mutations in the spike protein (98.5%) overlap with at least one known T-cell epitope. The influence on CD8+ T cell epitope generation by different HLA alleles was investigated for the three common mutations L5F, D614G and G1124V [49]. These mutations were predicted to result in epitope gains, losses or higher or lower HLA binding affinities. Our analysis suggests additional epitope-mutation pairs, which might result in a loss of the epitope and a chance of immune escape. All these findings demonstrate that SARS-CoV-2 mutants need to be set in the context of immune recognition to evaluate their implications for the global spreading of the pandemic and future preventive or therapeutic approaches in a timely manner.

## Conclusion and outlook

Human infections with SARS-CoV-2 are spreading globally since the beginning of 2020, necessitating preventive or therapeutic strategies and first steps towards an end to this pandemic were done with the approval of the first mRNA and vector based vaccines against SARS-CoV-2. Here, we show different types of variants (recurrent vs. individual, clonal vs. subclonal, hitting T-cell epitope vs. not-hitting) that can be incorporated in global efforts to sustainably prevent or treat infections. The underlying computational strategy might serve as a template for a platform to constantly analyze globally available sequencing data. In combination with a web-based platform to administer the results, this could help further guiding global vaccine design efforts to overcome the threats of this pandemic also in the future. In addition,

the results might serve as a starting point for further study of viral in-vivo evolution via tracking of subclonal variants and their co-occurrence in individual samples.

The importance of our approach is underlined by the emergence of SARS-CoV-2 lineages like the UK lineage B.1.1.7 [50], which is characterized by the accumulation of 17 variants; eight of those are located in the spike protein. This lineage has a higher transmissibility compared to other lineages [51]. The occurrence of this lineage questioned the efficacy of current vaccines, but first results showed that it at least unlikely will escape BNT162b-induced protection [52]. Interestingly, the individual variants can be traced back to samples from February (P681H, T716I, N501Y, A570D, S982A, and D1118H) and April (N501Y, A570D) of 2020. It needs to be mentioned that the available data, although representing a large cohort, might not reflect the real distribution of the circulating variants as mostly samples of specific interest will be sequenced. International sequencing efforts, combined data analysis and prediction of variant impact will be important tools for the future in order to ensure an early detection of such genomic variants of concern.

## Supporting information

**S1 Fig. Number and origin of publicly available SARS-CoV-2 sequence data over time.** The histogram shows the number of SARS-CoV-2 assembly sequences deposited at GISAID and NGS data deposited at SRA. Color coding indicates the sample origin. Countries summarized as "other" include: Afghanistan, Albania, Algeria, Andorra, Angola, Antigua and Barbuda, Argentina, Armenia, Aruba, Australia, Austria, Azerbaijan, Bahamas, Bahrain, Bangladesh, Barbados, Belarus, Belgium, Belize, Benin, Bermuda, Bolivia, Plurinational State of, Bonaire, Sint Eustatius and Saba, Bosnia and Herzegovina, Botswana, Brunei Darussalam, Bulgaria, Burkina Faso, Cambodia, Cameroon, Canada, Cayman Islands, Chile, Colombia, Comoros, Congo, Costa Rica, Croatia, Cuba, Cyprus, Czechia, Dominican Republic, Ecuador, Egypt, El Salvador, Equatorial Guinea, Estonia, Eswatini, Ethiopia, Faroe Islands, Finland, French Guiana, French Polynesia, Gabon, Gambia, Georgia, Ghana, Gibraltar, Guam, Guatemala, Guinea, Hong Kong, Hungary, Iceland, Indonesia, Iran, Islamic Republic of, Iraq, Ireland, Israel, Italy, Jamaica, Japan, Jordan, Kazakhstan, Kenya, Kuwait, Latvia, Lebanon, Lesotho, Liechtenstein, Lithuania, Luxembourg, Madagascar, Malawi, Malaysia, Mali, Malta, Martinique, Mauritius, Mexico, Moldova, Republic of, Monaco, Mongolia, Montenegro, Morocco, Mozambique, Myanmar, Nepal, Netherlands, New Zealand, Nigeria, North Macedonia, Northern Mariana Islands, Norway, Oman, Pakistan, Palestine, State of, Panama, Papua New Guinea, Paraguay, Peru, Philippines, Portugal, Reunion, Romania, Russian Federation, Rwanda, Saint Barthelemy, Saint Kitts and Nevis, Saint Lucia, Saint Martin (French part), Saint Vincent and the Grenadines, Saudi Arabia, Senegal, Serbia, Sierra Leone, Singapore, Slovakia, Slovenia, South Africa, Sri Lanka, Suriname, Sweden, Switzerland, Taiwan, Province of China, Thailand, Togo, Trinidad and Tobago, Tunisia, Uganda, Ukraine, United Arab Emirates, unknown, Uruguay, Uzbekistan, Venezuela, Bolivarian Republic of, Vietnam, Virgin Islands, British, Zambia, Zimbabwe and unknown.
(TIF)

**S1 Table. Overview of the 5,472 distinct non-synonymous mutations in the spike protein of SARS-CoV-2 detected in genome assemblies and NGS data sets.**
(XLSX)

## Acknowledgments

We gratefully acknowledge the authors from the originating laboratories responsible for obtaining the specimens, as well as the submitting laboratories where the sequence data were generated and shared via GISAID, NCBI Virus or the ENA, on which this research is based.

## Author Contributions

**Conceptualization:** Barbara Schrörs, Martin Löwer, Ugur Sahin.

**Formal analysis:** Barbara Schrörs, Pablo Riesgo-Ferreiro, Patrick Sorn, Ranganath Gudimella, Thomas Bukur, Thomas Rösler.

**Investigation:** Barbara Schrörs, Ranganath Gudimella, Thomas Bukur, Thomas Rösler.

**Writing – original draft:** Barbara Schrörs, Ranganath Gudimella, Thomas Bukur.

**Writing – review & editing:** Barbara Schrörs, Ranganath Gudimella, Martin Löwer, Ugur Sahin.

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
