## [Decision Letter · Decision Letter 0]

11 May 2021

PONE-D-21-09862

Large-scale analysis of SARS-CoV-2 spike-glycoprotein mutants demonstrates the need for continuous screening of virus isolates

PLOS ONE

Dear Dr. Löwer,

Thank you for submitting your manuscript to PLOS ONE. After careful consideration, we feel that it has merit but does not fully meet PLOS ONE’s publication criteria as it currently stands. Therefore, we invite you to submit a revised version of the manuscript that addresses the points raised during the review process.

Your manuscript was reviewed by 2 experts in the field. Both identified several important problems in your submission. Please review the attached comments and provide point-by-point responses.

We look forward to receiving your revised manuscript.

Kind regards,

Yury E Khudyakov, PhD

Academic Editor

PLOS ONE

Journal Requirements:

[The author(s) received no specific funding for this work.].    

We note that one or more of the authors are employed by a commercial company: TRON gGmbH and BioNTech SE

Reviewers' comments:

Reviewer's Responses to Questions

**Comments to the Author**

1. Is the manuscript technically sound, and do the data support the conclusions?

Reviewer #1: Partly

Reviewer #2: Yes

2. Has the statistical analysis been performed appropriately and rigorously? 

Reviewer #1: N/A

Reviewer #2: Yes

3. Have the authors made all data underlying the findings in their manuscript fully available?

Reviewer #1: Yes

Reviewer #2: Yes

4. Is the manuscript presented in an intelligible fashion and written in standard English?

Reviewer #1: Yes

Reviewer #2: Yes

5. Review Comments to the Author

Reviewer #1: Schrörs and colleagues's manuscript titled “Large-scale analysis of SARS-CoV-2 spike-glycoprotein mutants demonstrates the need for continuous screening of virus isolates” analyzed 146,917 SARS-CoV-2 genome assemblies and 2,393 NGS datasets from GISAID, NCBI sing bioinformatics tools. The data were outdated and the take-home messages are unclear.

1、 Bioinformatics are very useful tools to predict important variants. While the NISAID analysis are now very thorough, the re-analyze should provide some new clues for biologists to reference, but not just the re-listing of data.

2、 The data analyzed here are mostly up to Sep 2020, while the sequences are increased several times since then. The update of the data is recommended.

3、 The idea to analyze the individual subclone is good, however, the conclusion is deficient. How about the frequency of subclonal variants happened in other viruses? The comparison of SARS-CoV-2 to other virused may provide more use information.

4、 The effect of spike mutations on antibody and T cell recognition is definitely the key point, however, the data here is rough and the mutations mentioned in Fig4A are not including the most important mutations that already been proved. Therefore, the reliability of the method used here need to be verified.

Reviewer #2: Thank you for your hard work from sequence analysis and structure for mutation analysis to SASA analysis and T cell epitope analysis.

1. There is a place where English is awkward. The'of' between lines 66 and 67 is used twice.

Even on line 67, the sentence that is awkward in the English expression needs to be corrected.

2. In line 97, what the following reference would needed. Please review it and add it if appropriate.

Wu, F., Zhao, S., Yu, B. et al. A new coronavirus associated with human respiratory disease in China. Nature 579, 265--269 (2020). https://doi.org/10.1038/s41586-020-2008-3

3. Check if it is correct to capitalize PDB '6VXX' on line 112, and this form is'closed', which is difficult to see as a single form that is good for analyzing the coupling between RDB and Spike.

'Open'(up) type PDB was also analyzed, so please add it.

4. As one of the most common mutant forms, D614G written in result (around line 134) , which seems particularly important for open conformation analysis.

The SARS-CoV-2 Spike variant D614G favors an open conformational state

BY RACHAEL A. MANSBACH, SRIRUPA CHAKRABORTY, KIEN NGUYEN, DAVID C. MONTEFIORI, BETTE KORBER, S. GNANAKARAN

SCIENCE ADVANCES16 APR 2021: EABF3671

Please add a separate process for this or an explanation of the current data processing method.

5. In the analysis of the results, in spite of the many mutations, it is described that there are few mutations that cause significant conformation changes, and in the conclusion that nonetheless, it is argued that continuous screening is necessary. Claims that are not important may appear alternately and confusing, so if you can please write the part a little more refined form.

6. Please enhance the fig. esp. Fig1. and fig2.

And please transform the legend in plot more intuitive.

6. PLOS authors have the option to publish the peer review history of their article (what does this mean?). If published, this will include your full peer review and any attached files.

Reviewer #1: No

Reviewer #2: No

---

## [Author Response · Author response to Decision Letter 0]

28 Jul 2021

Response to Reviewers

Reviewer #1: Schrörs and colleagues's manuscript titled “Large-scale analysis of SARS-CoV-2 spike-glycoprotein mutants demonstrates the need for continuous screening of virus isolates” analyzed 146,917 SARS-CoV-2 genome assemblies and 2,393 NGS datasets from GISAID, NCBI sing bioinformatics tools. The data were outdated and the take-home messages are unclear.

1、 Bioinformatics are very useful tools to predict important variants. While the NISAID analysis are now very thorough, the re-analyze should provide some new clues for biologists to reference, but not just the re-listing of data.

Answer: We 100% percent agree that relisting existing data does not provide any benefit to researchers. The novelty of our analysis lies in the comprehensive detection of co-occurring variants, sub-clonal mutations and the impact of the mutations on T-cell immunogenicity of epitopes. All findings can be a valuable starting point for further experiments.

2、 The data analyzed here are mostly up to Sep 2020, while the sequences are increased several times since then. The update of the data is recommended.

Answer: Good point – we massively increased the scope of the analysis and now include over 10^6 samples and the respective datasets up to a collection date of April 2021.

3、 The idea to analyze the individual subclone is good, however, the conclusion is deficient. How about the frequency of subclonal variants happened in other viruses? The comparison of SARS-CoV-2 to other virused may provide more use information.

Answer: We agree that such a comparison might be useful. However, the detected subclones a relatively rare and the detection itself is only reliable because of the massive amount of available NGS data for SARS-CoV-2, which is not true for any other virus. Rather we recommend further experimental study of subclonal events and added this point to the conclusion of the manuscript.

4、 The effect of spike mutations on antibody and T cell recognition is definitely the key point, however, the data here is rough and the mutations mentioned in Fig4A are not including the most important mutations that already been proved. Therefore, the reliability of the method used here need to be verified.

Thanks for pointing this out. After careful consideration, we concluded that the structure-based approach of determining potential effects on antibody binding was not well thought out and indeed lacks validation. We replaced this analysis by a more detailed and advanced analysis of the impact of the variants on HLA-epitiope binding, which is a commonly used method in computational cancer immunology. We cite proper references in order to demonstrate the validity. 

Reviewer #2: Thank you for your hard work from sequence analysis and structure for mutation analysis to SASA analysis and T cell epitope analysis.

1. There is a place where English is awkward. The'of' between lines 66 and 67 is used twice.

Even on line 67, the sentence that is awkward in the English expression needs to be corrected.

Answer: Thanks – we corrected this error.

2. In line 97, what the following reference would needed. Please review it and add it if appropriate.

Wu, F., Zhao, S., Yu, B. et al. A new coronavirus associated with human respiratory disease in China. Nature 579, 265--269 (2020). https://doi.org/10.1038/s41586-020-2008-3

Answer: Thanks, we added this reference.

3. Check if it is correct to capitalize PDB '6VXX' on line 112, and this form is'closed', which is difficult to see as a single form that is good for analyzing the coupling between RDB and Spike.

'Open'(up) type PDB was also analyzed, so please add it.

4. As one of the most common mutant forms, D614G written in result (around line 134) , which seems particularly important for open conformation analysis.

The SARS-CoV-2 Spike variant D614G favors an open conformational state

BY RACHAEL A. MANSBACH, SRIRUPA CHAKRABORTY, KIEN NGUYEN, DAVID C. MONTEFIORI, BETTE KORBER, S. GNANAKARAN

SCIENCE ADVANCES16 APR 2021: EABF3671

Please add a separate process for this or an explanation of the current data processing method.

5. In the analysis of the results, in spite of the many mutations, it is described that there are few mutations that cause significant conformation changes, and in the conclusion that nonetheless, it is argued that continuous screening is necessary. Claims that are not important may appear alternately and confusing, so if you can please write the part a little more refined form.

Answer to comments 3, 4 and 5: After careful consideration, we concluded that the structure-based approach of determining potential effects on antibody binding was not well thought out and indeed lacks validation. We replaced this analysis by a more detailed and advanced analysis of the impact of the variants on HLA-epitiope binding, which is a commonly used method in computational cancer immunology. We cite proper references in order to demonstrate the validity. The results of this analysis demonstrate, that the mutant form of an epitope has the potential to disrupt epitope recognition and that the continued data analysis of NGS data in this regard can detect such events early.

6. Please enhance the fig. esp. Fig1. and fig2.

And please transform the legend in plot more intuitive.

Answer: We enlarged subplots of Fig 1 and Fig 2, as well as improved the description of the Figures in the legends.

---

## [Editor Report · Decision Letter 1]

2 Aug 2021

Large-scale analysis of SARS-CoV-2 spike-glycoprotein mutants demonstrates the need for continuous screening of virus isolates

PONE-D-21-09862R1

Dear Dr. Löwer,

We’re pleased to inform you that your manuscript has been judged scientifically suitable for publication and will be formally accepted for publication once it meets all outstanding technical requirements.

Kind regards,

Yury E Khudyakov, PhD

Academic Editor

PLOS ONE
---

## [Editor Report · Acceptance letter]

17 Sep 2021

PONE-D-21-09862R1 

Large-scale analysis of SARS-CoV-2 spike-glycoprotein mutants demonstrates the need for continuous screening of virus isolates 

Dear Dr. Löwer:

I'm pleased to inform you that your manuscript has been deemed suitable for publication in PLOS ONE. Congratulations! Your manuscript is now with our production department. 

Kind regards, 

on behalf of

Dr. Yury E Khudyakov 

Academic Editor

PLOS ONE